# Hygroscopic holey graphene aerogel fibers enable highly efficient moisture capture, heat allocation and microwave absorption

Yinglai Hou[1,2,4], Zhizhi Sheng[2,4], Chen Fu[2], Jie Kong [1✉] & Xuetong Zhang [2,3✉]

Aerogel fibers have been recognized as the rising star in the fields of thermal insulation and wearable textiles. Yet, the lack of functionalization in aerogel fibers limits their applications. Herein, we report hygroscopic holey graphene aerogel fibers (LiCl@HGAFs) with integrated functionalities of highly efficient moisture capture, heat allocation, and microwave absorption. LiCl@HGAFs realize the water sorption capacity over $4.15\,g\,g^{-1}$, due to the high surface area and high water uptake kinetics. Moreover, the sorbent can be regenerated through both photo-thermal and electro-thermal approaches. Along with the water sorption and desorption, LiCl@HGAFs experience an efficient heat transfer process, with a heat storage capacity of $6.93\,kJ\,g^{-1}$. The coefficient of performance in the heating and cooling mode can reach 1.72 and 0.70, respectively. Notably, with the entrapped water, LiCl@HGAFs exhibit broad microwave absorption with a bandwidth of 9.69 GHz, good impedance matching, and a high attenuation constant of 585. In light of these findings, the multifunctional LiCl@HGAFs open an avenue for applications in water harvest, heat allocation, and microwave absorption. This strategy also suggests the possibility to functionalize aerogel fibers towards even broader applications.

[1] Shaanxi Key Laboratory of Macromolecular Science and Technology, School of Chemistry and Chemical Engineering, Northwestern Polytechnical University, 710072 Xi'an, PR China. [2] Suzhou Institute of Nano-Tech and Nano-Bionics, Chinese Academy of Sciences, 215123 Suzhou, PR China. [3] Division of Surgery & Interventional Science, University College London, London NW3 2PF, UK. [4]These authors contributed equally: Yinglai Hou, Zhizhi Sheng. ✉email: kongjie@nwpu.edu.cn; xtzhang2013@sinano.ac.cn

Aerogel fibers have been the focal points in a broad spectrum of applications, ranging from thermal insulation[1–4], wearable textiles[3,5], to stimuli-responsive electronics[6], due to their high specific area, high porosity, low density, and low thermal conductivity. A variety of materials can be fabricated into aerogel fibers, such as polymers (e.g., Kevlar, polyimide)[3,7], ceramics (e.g., silica, boron nitride)[1,4,8,9], carbon-based materials (e.g., CNT, graphene, MXene)[6,10–12], and hybrid materials (e.g., cellulose/cobalt ferrite, silk fibroin/graphene oxide, graphene/Ni)[5,13,14]. Different strategies are exploited for the construction of aerogel fibers starting from variant nanoscale building blocks, for instance, reaction spinning[4], coaxial spinning[5], wet spinning[6,10], and sol-gel confined transition method[7]. With further freeze-drying or supercritical $CO_2$ drying, aerogel fibers with superior thermal insulation[3,4], superhydrophobicity[3,4], high transparency[4], high mechanical strength, and desirable flame-resistance[7] can be realized, depending on the selection of nanoscale building blocks. The resulting aerogel fibers can be easily knotted, bent, and even woven into fabrics for wearable applications at room temperature or under extreme environments[3]. The low thermal conductivity of aerogel materials, combined with the shape flexibility of one-dimensional fibers, has facilitated the development of thermal insulation textiles.

However, to extend the applications beyond thermal insulation, the functionalization of aerogel fibers is quite necessary. Specifically, the porous architecture in aerogel fibers provides sufficient confinements and high surface area to host other guest components for their further functionalization[15]. That is, aerogel fibers possess a high surface area that can bear rich functionalities to interact with ions or molecules[7,11,16], and even external stimuli[6]. Prior efforts have been focused on the infiltration of an intelligent guest into aerogel fibers. For example, graphene aerogel fibers have been adopted as the porous host to incorporate the phase change materials, exhibiting self-clean super-hydrophobic surface and excellent multiple responsiveness to external stimuli (electric field/light/thermal field) as well as reversible energy storage and conversion capability[6]. Introducing the phase change material into aerogel fibers would also improve the thermal comfort of humans by adjusting the microenvironment through thermal storage and release of phase change materials[3]. However, current aerogel fibers are still limited on a single function, multiple functionalities are greatly needed when encountering variant or complex environments. Therefore, it is highly desirable to develop multifunctional aerogel fibers for broader promising applications.

Herein, we report the hygroscopic holey graphene aerogel fibers (LiCl@HGAFs), where the holey graphene aerogel fibers host efficient hygroscopic salt LiCl, enabling superior moisture capture, heat allocation, and microwave absorption performance. The holey graphene aerogel porous matrix provides not only sufficient binding sites and surface area for water uptake but also abundant water transport pathways through the etched nanopores. These LiCl@HGAFs exhibit 4.15 g g$^{-1}$ moisture sorption capacity at 90% relative humidity and multiple pathways to perform sorption/desorption processes, and thermodynamics and kinetics of the water sorption with the LiCl@HGAFs are determined. In addition, during the process of water sorption and desorption, not only mass transfer but also heat transfer occurs. Benefitting from moisture sorption performance, LiCl@HGAFs are also demonstrated as adsorption-based heat transfer (AHT) devices such as adsorption-driven cooling/chiller and adsorption-driven heat pump. Bearing water as the ideal working fluid (the latent heat of evaporation up to 44 kJ mol$^{-1}$ at room temperature and recyclability), AHT devices have the potential to be environmental-friendly, non-flammable, and low-cost, targeting a drastic reduction of energy consumption for cooling and heating owing to the potential use of natural solar energy or waste heat from industrial plants. Furthermore, the LiCl@HGAFs contain above 2 g g-1 saline water in the porous host after absorbing the moisture, showing microwave absorption performance. The microwave absorption of the LiCl@HGAFs is greatly improved after the sorption of water. The effect adsorption bandwidth has been improved from 0 to 9.69 GHz (8.31–18 GHz). Hence, the LiCl@HGAFs hold great promise for water harvest from the air, sorption-based heat allocation, as well as microwave absorption, paving the way towards multifunctional fiber-based devices and emerging applications.

## Results

**Fabrication strategy and functional design.** The design strategy for LiCl@HGAFs is illustrated in Fig. 1. LiCl is selected as the active salt decorated within the fiber because of its low density, low dehydration temperature, and super high water sorption capacity[17]. First, the holey graphene oxide (HGO) was prepared by etching graphene oxide (GO) in $H_2O_2$ at 100 °C followed by washing and centrifuging. Then, the LiCl@HGAFs were fabricated by wet spun, reduction, supercritical drying (Sc-drying), and filling with LiCl in sequence (Fig. 1a). LiCl@HGAFs are further demonstrated with outstanding water harvest, heat allocation, and microwave adsorption behavior (Fig. 1b). Water molecules can be captured by LiCl@HGAFs and easily released based on the electro-thermal or photo-thermal effect of graphene. Efficient heat allocation was realized since heat is reversibly generated and released along with the adsorption and desorption of water. With water saturated in the interconnected porous structure, LiCl@HGAFs also show outstanding microwave adsorption performance, due to dipole polarization of water molecules, dipole polarization at the nanopore defects within the graphene plane, and multiple reflections between two-dimensional (2D) graphene sheets.

During the etching process, the carbon atoms in the actively defective zones of GO can be oxidized by $H_2O_2$, thereby generating nanopores gradually (Fig. 2a)[18]. The transmission electron microscopy (TEM) study shows that abundant in-plane nanopores are produced on the GO sheets by etching with $H_2O_2$ (Fig. 2b(ii)). And as the etching time lengthens, the number of pores in the sheets increases (Supplementary Fig. 1b–d). As a contrast, no obvious in-plane nanopores are observed in the sample without $H_2O_2$ etching (Fig. 2b(i) and Supplementary Fig. 1a). The resulting HGO has fewer oxygen functionalities and defects than GO, which can be concluded from the decreased intensity ratio of the D band to G band in Raman spectra (Supplementary Fig. 2). There are two typical peaks in the spectra, where one peak at 1350 cm$^{-1}$ is attributed to the D band with disordered and defected structure and the other peak at 1583 cm$^{-1}$ corresponding to the G band with graphitic structure[19]. After etching, the $I_D/I_G$ decreases from 0.9937 to 0.9197, indicating that GO is deoxygenated during the etching process, which is consistent with previous reports[20,21]. It is proposed that oxidative-etching initiates and propagates in the oxygenic defect sites within the basal plane of GO, resulting in the removal of oxygenated carbon atoms and the formation of carbon vacancies that eventually extend into nanopores[21,22]. Further studying the oxygen functional groups on HGO sheets by XPS proves that the C-O group (~286 eV) is decreased after $H_2O_2$ etching (Supplementary Fig. 3). After centrifugation, the HGO aqueous lyotropic crystal phase suspension is analogous to that of GO suspension with a high concentration, which can be observed under a polarizing microscope (Fig. 2c).

By injecting HGO suspension with a spinning nozzle of 500 μm into $CaCl_2$ aqueous solution, the HGO hydrogel fibers were

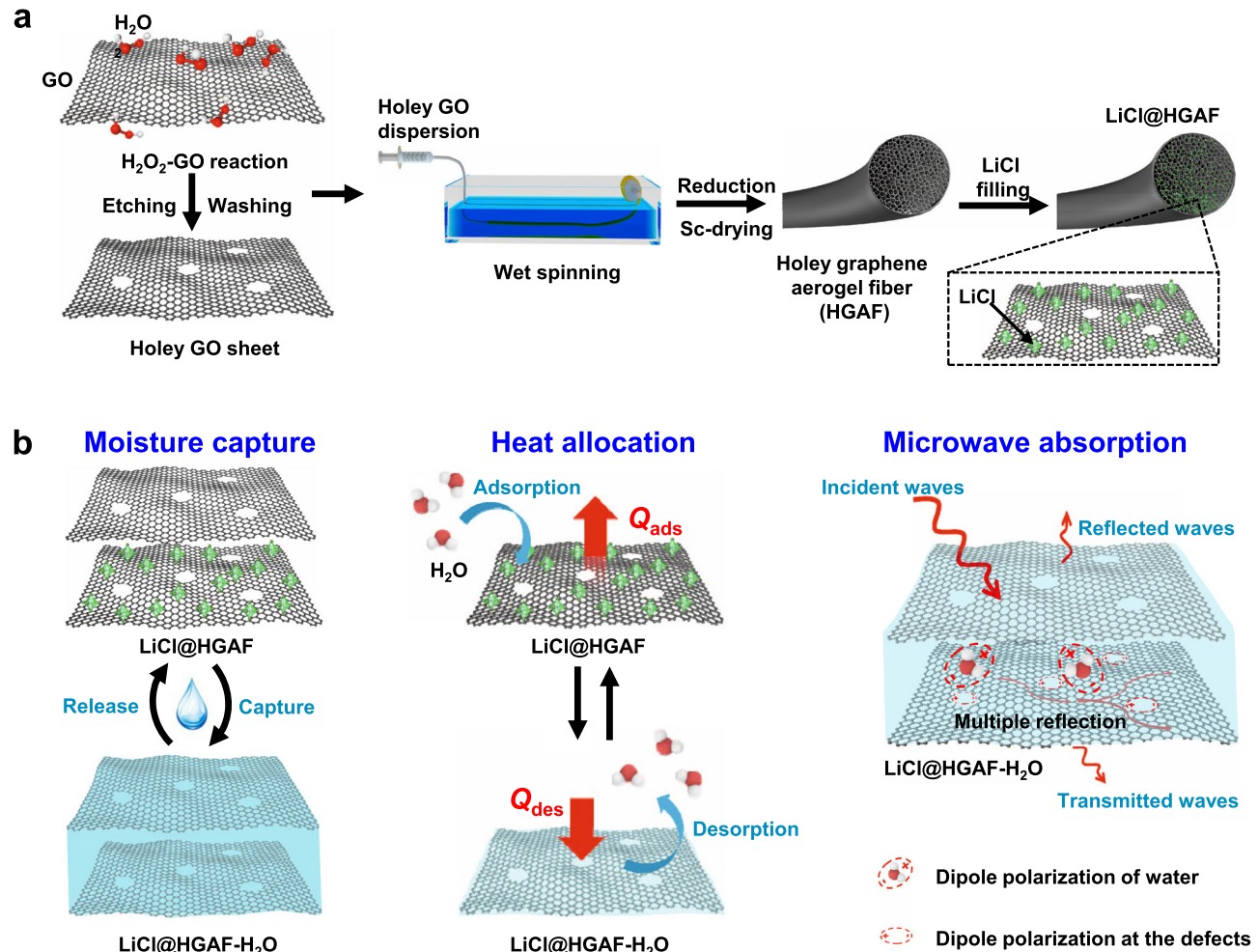

**Fig. 1 Fabrication strategy and application of LiCl@HGAFs. a** Schematic illustration of the fabrication of hygroscopic holey graphene aerogel fibers (LiCl@HGAFs). HGAFs were obtained by wet spinning, HI reduction, and supercritical drying. LiCl was introduced by simple impregnation. **b** Schematic illustrations on moisture capture, heat allocation, and microwave absorption, respectively. LiCl in the fiber can effectively capture moisture and liquefy it, which can be regenerated by heating and harvested water. The adsorption heat is generated in the adsorption process ($Q_{ads}$) and the heat is absorbed in the regeneration process ($Q_{des}$). This heat transfer property makes the LiCl@HGAFs applicable in heat storage and heat distribution allocation. In the process of moisture capture, the liquid water produced in the fibers has a significant loss effect on microwave adsorption, which can significantly improve the microwave absorption performance of the fibers.

obtained (Supplementary Fig. 4). Compared with GO hydrogel fibers, HGO hydrogel fibers have a darker color, indicating a lower degree of oxidation. The liquid crystals of HGO could be self-assembled by the shear force applied through the syringe piston and $Ca^{2+}$ offered interlayer and intralayer cross-linking bridges between the oxygen-containing groups to improve the stability of HGO hydrogel fibers. Hydriodic acid (HI) was employed to reduce the HGO hydrogel fiber. The XPS result shows that -C-O and C = O groups are decreased after reduction (Supplementary Fig. 3 and Supplementary Fig. 5b). However, compared with GAF, holey graphene aerogel fiber (HGAF) has more oxygen and less reduction (Supplementary Fig. 5a), which is attributed to more defects. Raman spectra show that obtained HGAFs have more defects than GAFs, the $I_D/I_G$ of HGAFs is 1.13–1.23, higher than the value for GAFs ($I_D/I_G = 1.06$) (Supplementary Fig. 6). This may be due to the fact that more defects are produced in the etching process, and the oxygen-containing groups at the defects are more difficult to reduce. However, the existence of these defects will affect the final properties of the fiber. The electrical conductivity of the fiber decreases significantly with the increase of defects (Supplementary Fig. 7). In the Sc-drying process, the solvent has a low surface

tension, and the transformation from wet gel to aerogel is completed with a contraction ratio of 0.42 by preserving the porous skeleton. The hierarchical structure of obtained HGAFs with the density of 0.23 g cm$^{-3}$ was revealed by the field emission electron microscopy (FESEM). The images indicate that holey graphene sheets are assembled in uniform long-range order as same as GAF (Fig. 2g, h and Supplementary Fig. 8), which are induced by the shearing force through the injection. Due to hydrogen peroxide etching, the pore size distribution and pore volume of the fiber are significantly increased (Supplementary Fig. 9). The specific surface area of HGAFs (356.3 m$^2$ g$^{-1}$) is higher than GAF (218.8 m$^2$ g$^{-1}$), which can be attributed to the existence of nanopores on the graphene sheets. Furthermore, The HGAF possesses flexibility (bending stiffness $R_f = 3.08 \times 10^{-9}$ N m$^2$) and can be knotted or woven into a textile (Fig. 2d–f). Additionally, in view of the aging effect on the preparation of aerogel materials, the specific surface area, electrical conductivity, tensile strength, and average diameter of HGAFs with different aging times (1, 2, and 3 days) were investigated in Supplementary Figs. 10–13. The specific surface area decreases with increasing aging time (Supplementary Fig. 10). It can be attributed to the fact that the diffusion of ions during aging increases the ionic

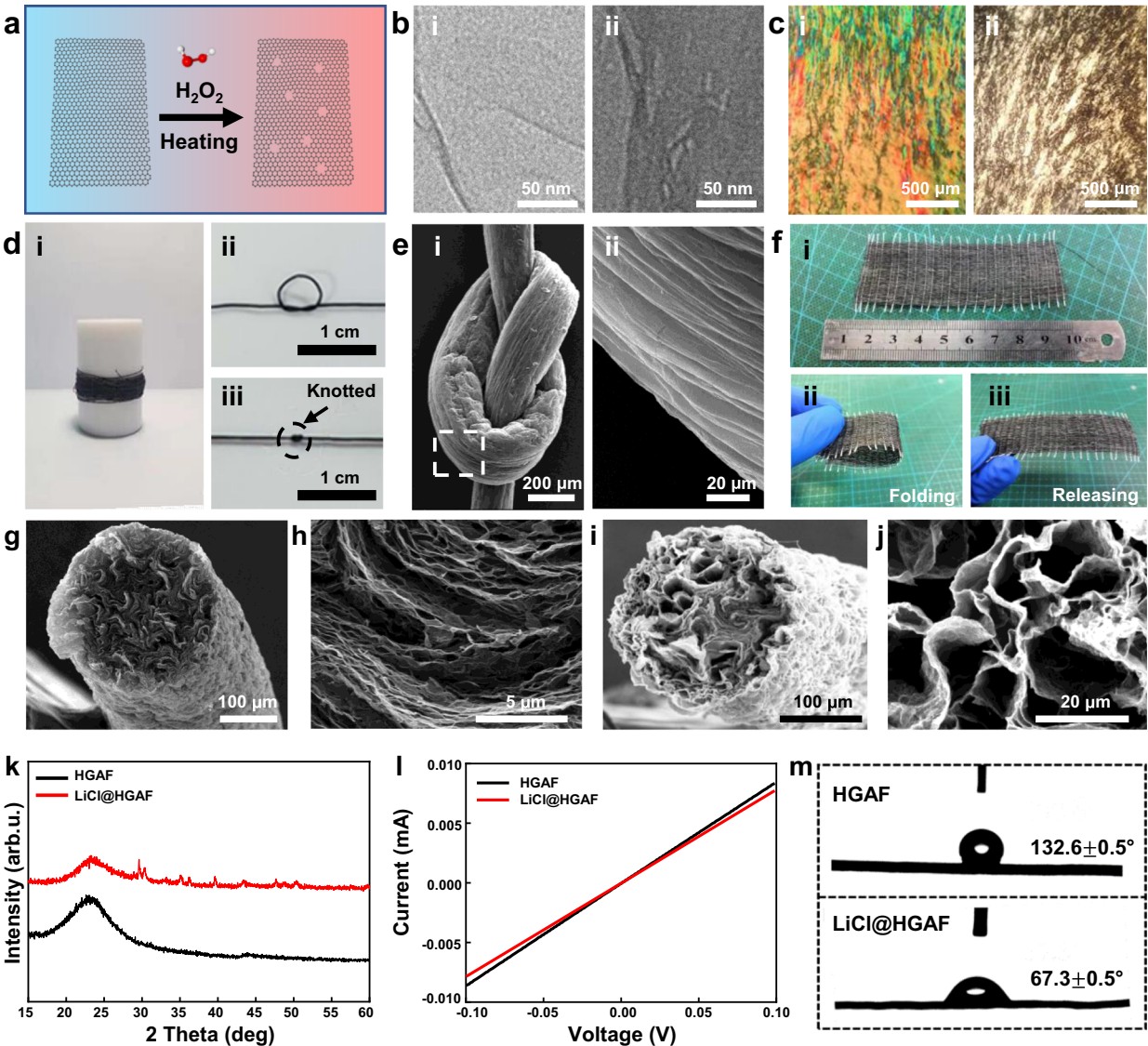

**Fig. 2 Characterization of LiCl@HGAFs. a** Schematic of the nanopores on graphene oxide formation by $H_2O_2$ etching. **b** TEM images of GO (i) and HGO (ii) sheets. **c** The optical images of GO liquid crystal (i) and HGO liquid crystal (ii) made from etching GO by $H_2O_2$. **d** A digital photo of HGAF (i) and knotted HGAF (ii). **e** SEM images of the knotted holey graphene aerogel fiber (i) and a partial enlargement of the knot (ii). **f** Photographs of HGAF based textile (i) and folding test (ii–iii). **g**, **h** SEM images of the holey graphene aerogel fiber. **i**, **j** SEM images of LiCl@HGAF-3 (3 wt.% LiCl). **k** XRD patterns of HGAFs and LiCl@HGAFs. Source data are provided as a Source Data file. **l** I-E curves of HGAF and LiCl@HGAF. Source data are provided as a Source Data file. **m** The water contact angle measurements for HGAF and LiCl@HGAF.

crosslinks between holey graphene sheets, making them more tightly packed and causing more shrinkage (Supplementary Fig. 13, the average diameter of the fibers decreases from 329.41 μm to 273.44 μm with aging time). Nevertheless, at the same time, the electrical conductivity (from 146.77 S m$^{-1}$ to 211.69 S m$^{-1}$, Supplementary Fig. 11) and tensile strength (from 0.77 MPa to 1.03 MPa, Supplementary Fig. 12) of fibers increase with the aging time. Moreover, we employ a liquid impregnation strategy for the construction of the LiCl@HGAFs. Obviously, the hygroscopic LiCl is uniformly distributed across the graphene sheets (Fig. 2i, j and Supplementary Fig. 14). The distribution of Cl element derived from LiCl further reveals that LiCl crystals are homogeneously anchored on the holey graphene sheets (Supplementary Fig. 15). With the increase of LiCl mass loading (3 wt.%, 5 wt.%, and 7 wt.%, denoted as LiCl@HGAF-3, LiCl@HGAF-5, and LiCl@HGAF-7, respectively), it can be observed that the filling amount in the porous architecture increased significantly.

When the loading amount is high enough, LiCl is prone to decorate around the whole graphene sheets in the fiber. X-ray Diffraction (XRD) patterns of HGAFs, LiCl@HGAF show the structure of HGAF is preserved after the impregnation of LiCl (Fig. 2k). The newly appeared diffraction peak in LiCl@HGAF is assigned to lithium chloride hydrate, indicating that water molecules can rapidly interact with the LiCl to form LiCl·$H_2O$, which is known as hydration reaction[17,23]. The electrical conductivity of LiCl@HGAF is illustrated by the I-E curve (Fig. 2l), where the conductivity of LiCl@HAGF (149 S m$^{-1}$) drops only slightly in comparison with that of HGAF (163 S m$^{-1}$). This reveals that the impregnation and subsequent freeze-drying do not damage the three-dimensional network structure in the fiber, and the slight decrease in conductivity is attributed to the existence of non-conductive hygroscopic salt crystals on the graphene sheets. Besides, HGAF is inherently hydrophobic with the contact angle of 132.6 ± 0.5°, while LiCl@HGAF is

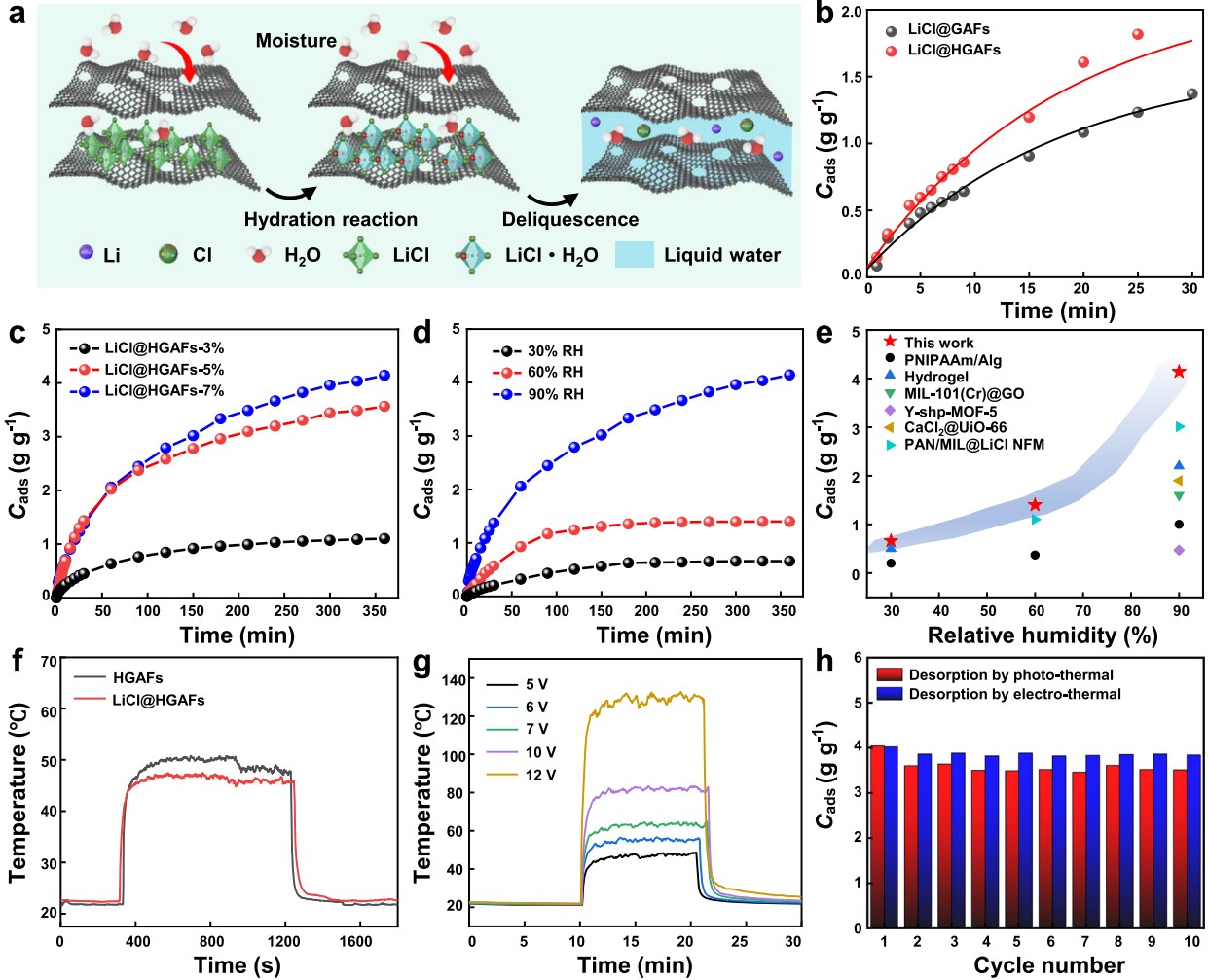

**Fig. 3 High efficient moisture capture by LiCl@HGAFs. a** Schematic illustration of the moisture sorption process. In this process, LiCl reacts with water first to form LiCl·$H_2O$, and then deliquesce to form LiCl solution, and the captured moisture exists in the form of liquid water (blue-shaded region). **b** Water uptake of LiCl@GAFs and LiCl@HGAFs at 25 °C and 90% RH in 30 min. Source data are provided as a Source Data file. **c** Water uptake of LiCl@HGAFs with different salt contents at 25 °C and 90% RH. Source data are provided as a Source Data file. **d** Water uptake of LiCl@HGAFs under the relative humidity of 30%, 60%, and 90%. Source data are provided as a Source Data file. **e** Comparison of water sorption capacity with reported hygroscopic materials: PNIPAAm/Alg[54], Hydrogel[28], MIL-101(Cr)@GO[37], CaCl$_2$@UiO-66[55], Y-shp-MOF-5[42], and PAN/MIL@LiCl NFM[56]. The materials marked in the blue-shaded region are LiCl@HGAFs. Source data are provided as a Source Data file. **f** Temperature response of HGAFs and LiCl@HGAFs under one-sun irradiation. Source data are provided as a Source Data file. **g** Temperature response of LiCl@HGAFs under various voltages. Source data are provided as a Source Data file. **h** Cycling stability of the sorption-desorption process of LiCl@HGAFs under photo-thermal or electro-thermal conditions. Source data are provided as a Source Data file.

hydrophilic with the contact angle of 67.3 ± 0.5°, supporting that the presence of salt causes the fiber to change from hydrophobic to hydrophilic (Fig. 2m and Supplementary Fig. 16). The hydrophilicity of the composite fiber also guarantees the ease of water sorption.

**Moisture sorption and desorption behaviors of LiCl@HGAFs.** Figure 3 displays the moisture sorption mechanism and moisture capture characterization of LiCl@HGAFs. The whole water uptake process includes three-step sorption processes: (i) the chemisorption from LiCl@HGAFs to LiCl@HGAFs·$H_2O$, (ii) the deliquescence of LiCl·$H_2O$ to LiCl concentrated solution, and (iii) solution absorption from the concentrated solution to dilute solution (Fig. 3a). The saturated solution formed continues to capture water molecules from the air to form a diluted solution, demonstrating the absorption behavior of the liquid absorbent in this condition. The moisture sorption performance of LiCl@H-GAFs was gravimetrically evaluated at 25 °C. Compared with

LiCl@GAFs, LiCl@HGAFs exhibits faster adsorption kinetics during the same period (Fig. 3b, LiCl@HGAFs: 1.81 g g⁻¹, 30 min; LiCl@GAFs: 1.37 g g⁻¹, 30 min), due to the presence of etched nanopores on the sheet that increase the diffusion pathways for water molecules. However, with increasing the sorption time up to 350 min, two curves would gradually approach to each other (Supplementary Fig. 17), originating from the deliquescence of LiCl·$H_2O$ inside the fiber and that LiCl aqueous solution gradually fills the entire fiber.

Combining the various pore structure and high porosity of the aerogel matrix with the strong moisture sorption of LiCl, LiCl@HGAFs show excellent moisture sorption capacity. Meanwhile, the LiCl content in LiCl@HGAF plays a crucial role in the moisture sorption capacity. As shown in Fig. 3c, at the temperature of 25 °C and the humidity of 90% RH, the moisture sorption capacity increases dramatically with increasing the loading fraction of LiCl. LiCl@HGAF-7% and LiCl@HGAF-5% exhibit much higher sorption capacity than LiCl@HGAF-3%,

corresponding to $4.14 \, g\,g^{-1}$, $3.56 \, g\,g^{-1}$, and $1.11 \, g\,g^{-1}$, respectively. Interestingly, the difference between the sorption kinetics (i.e., the slope of the curve at the initial stage) of LiCl@HGAF-7% and LiCl@HGAF-5% is not obvious (Supplementary Fig. 18a), mainly due to the LiCl solution entrapped in the pores of the fiber. In addition, LiCl@HGAFs exhibit high moisture sorption capacity and kinetics under a wide range of humidities (Fig. 3d and Supplementary Fig. 18b). Although the decrease in relative humidity has a huge impact on the moisture absorption of the fiber, the moisture absorption can reach $1.4 \, g\,g^{-1}$ at moderate humidity (60% RH), with $0.66 \, g\,g^{-1}$ at lower humidity (30% RH). Compared with previously reported moisture sorption materials, LiCl@HGAFs in this work exhibit superior sorption capacity at 25 °C over a broad range of humidities, achieving more than 30% above the best material from some literature at 90% RH (Fig. 3e). Furthermore, dehumidification tests of a series of moisture sorption materials were conducted in a closed chamber with the same initial relative humidity (Supplementary Fig. 19). For the dehumidification performance of the sorbents with the same mass (Supplementary Fig. 19a), LiCl@HGAFs outperforms most of the compared sorbents and is slightly weaker than LiCl. For the sorbents with the same volume (Supplementary Fig. 19b), LiCl@HGAFs shows moderate hygroscopic performance compared with other sorption materials. As a result, LiCl@HGAF can fully meet the needs of practical applications and can be easily operated at ambient temperature, surpassing the performance of other granular or membraniform moisture sorbents.

LiCl@HGAFs possess both high solar-thermal conversion and high electro-thermal capability. Therefore, both solar energy and electrical energy can be used as the driving force to desorb the captured water from the sorbent. These regeneration processes for the moisture sorbents are environmental-friendly and meanwhile, low energy input is required. Both HGAFs and LiCl@HGAFs show a rapid photothermal response under one-sun radiation ($1 \, kW/m^2$), with the temperature rising from 22 °C to 46 °C and 44 °C within 44 s, respectively (Fig. 3f and Supplementary Fig. 21). The highest temperature reaches around 50 °C for HGAFs and 47 °C for LiCl@HGAFs, respectively. The desorption of LiCl@HGAFs was carried out under a typical water vapor pressure of 1.2 kPa. Considering that the desorption temperature of LiCl@HGAFs depends on the dehydration of LiCl·H$_2$O, the theoretical desorption temperature to the product of LiCl is 69 °C according to Clausius–Clapeyron equilibrium equation[24]. Therefore, under the photo-thermal condition (47 °C), LiCl@HGAFs undergo the desorption from LiCl solution to LiCl·H$_2$O, where the regeneration degree can reach 83.4% (Supplementary Fig. 20). Unlike granular sorbents, LiCl@HGAFs are featured with interconnecting conductive networks, so they can also be desorbed and regenerated by electric heating. The fibers show excellent electro-thermal performance with the temperature rising to 131 °C under 12 V (Fig. 3g and Supplementary Fig. 23), sufficient to enable the complete desorption of the sorbent. Moreover, the fiber was subjected to 10 times sorption-desorption cycles using photo-thermal desorption and electro-thermal desorption, respectively. After five cycles, the fiber sorption rate coefficient drops from $1.575 \times 10^{-4} \, s^{-1}$ to $1.393 \times 10^{-4} \, s^{-1}$ (Supplementary Fig. 18c and Supplementary Table 1), caused by the slight agglomeration of hygroscopic salt. The overall cycle stability of GAFs and HAGFs is maintained at a relatively stable level up to 10 cycles, both demonstrating rapid cycling capability of water capture and water release (Fig. 3h and Supplementary Fig. 22). The obtained transmission electron microscopy (TEM) images of LiCl@GAFs and LiCl@HGAFs before and after the cyclic test show that there is no obvious change in fiber structure before and after the cyclic test (Supplementary Fig. 24). The corresponding elemental maps of LiCl@HGAFs after cyclic test show that Cl element derived from LiCl was uniformly distributed along the holey graphene sheets along with C element, further revealing the satisfactory stability of LiCl@HGAFs (Supplementary Fig. 25).

The sorption kinetic performance can be expressed by the sorption rate coefficient, which can be determined by the following equation:[25]

$$\frac{x}{x_{eq}} = 1 - e^{-kt} \quad (1)$$

where, $x_{eq}$ stands for the equilibrium water sorption quantity (g g$^{-1}$), $x$ stands for the dynamic water sorption quantity (g g$^{-1}$), k stands for the sorption rate coefficient (s$^{-1}$), and t stands for sorption time (s). As the salt content increases, the sorption rate coefficient of the fiber decreases from $2.388 \times 10^{-4} \, s^{-1}$ for LiCl@HGAFs-3% to $1.653 \times 10^{-4} \, s^{-1}$ for LiCl@HGAFs-7% (Supplementary Fig. 18a and Supplementary Table 1). This is caused by the increased diffusion resistance due to the hygroscopic salt incorporated into the matrix pores[26]. In addition, with the increase of humidity, although the specific sorption capacity increases from $0.66 \, g\,g^{-1}$ (30% RH) to $4.14 \, g\,g^{-1}$ (90% RH), the sorption rate coefficient shows a downward trend (from $2.998 \times 10^{-4} \, s^{-1}$ at 30% RH to $1.653 \times 10^{-4} \, s^{-1}$ at 90% RH). This is because the hygroscopic salt will form a solution during the sorption process, which changes the sorption mechanism from solid adsorption to liquid absorption. As the relative humidity increases, the liquid absorption mechanism will be dominated, leading to a lower sorption rate coefficient or slower mass transfer rate. Still, the sorption rate coefficient of LiCl@HGAFs ($1.653 \times 10^{-4} \, s^{-1}$ to $2.998 \times 10^{-4} \, s^{-1}$) in this work is higher than that of silica gel impregnated with LiCl, LiBr, and CaCl$_2$, where the sorption rate coefficient ranges from $9.03 \times 10^{-5} \, s^{-1}$ to $1.49 \times 10^{-4} \, s^{-1}$ [27].

**Adsorption-based heat transfer application based on moisture sorption property**. As a multifunctional hygroscopic material, in addition to obtaining water from the air, it can also be used as a thermal energy storage material along with an excellent water sorption property. Adsorption-based heat transfer (AHT) devices, such as adsorption-driven cooling/chiller, adsorption-driven heat pumps, and thermal batteries, have been recently proposed as cutting edge renewable energy alternative solutions to meet the huge global energy demands for heating and cooling[28]. We further utilize the LiCl@HGAFs for heat allocation in AHT devices. A schematic working principle of AHT devices is illustrated in Fig. 4a. AHT system typically operates under a full cycle of water adsorption/desorption. Incorporating the efficient "adsorbent-water" working pair, AHT devices can operate through the endothermic process of water evaporation or exothermic process of water adsorption[29]. Adsorption-driven heating can thus operate with the released heat of adsorption ($Q_{ads}$) and condensation ($Q_{cond}$). To evaluate our LiCl@HGAFs as sorbents for thermal batteries, we selected LiCl@HGAF-7 because it has a higher working capacity due to the high salt content. The water-sorption behavior of LiCl@HGAF-7 was explored at four different temperatures (Fig. 4b). The isotherm curves obtained at four different temperatures (293 K, 303 K, 313 K, and 323 K) are nearly linear at moderate pressures from 0.15 to 0.7 and can therefore be described by the Freundlich model and S-B-K model (Supplementary Fig. 26, Supplementary Tables 2 and 3)[30,31].

The variable-temperature water vapor isotherms for LiCl@H-GAFs are also used to validate a characteristic curve. Two characteristic curves under 303 K and 313 K are close, indicating that the characteristic curve is temperature invariant, justifying its use to calculate isotherms at other temperatures (Supplementary

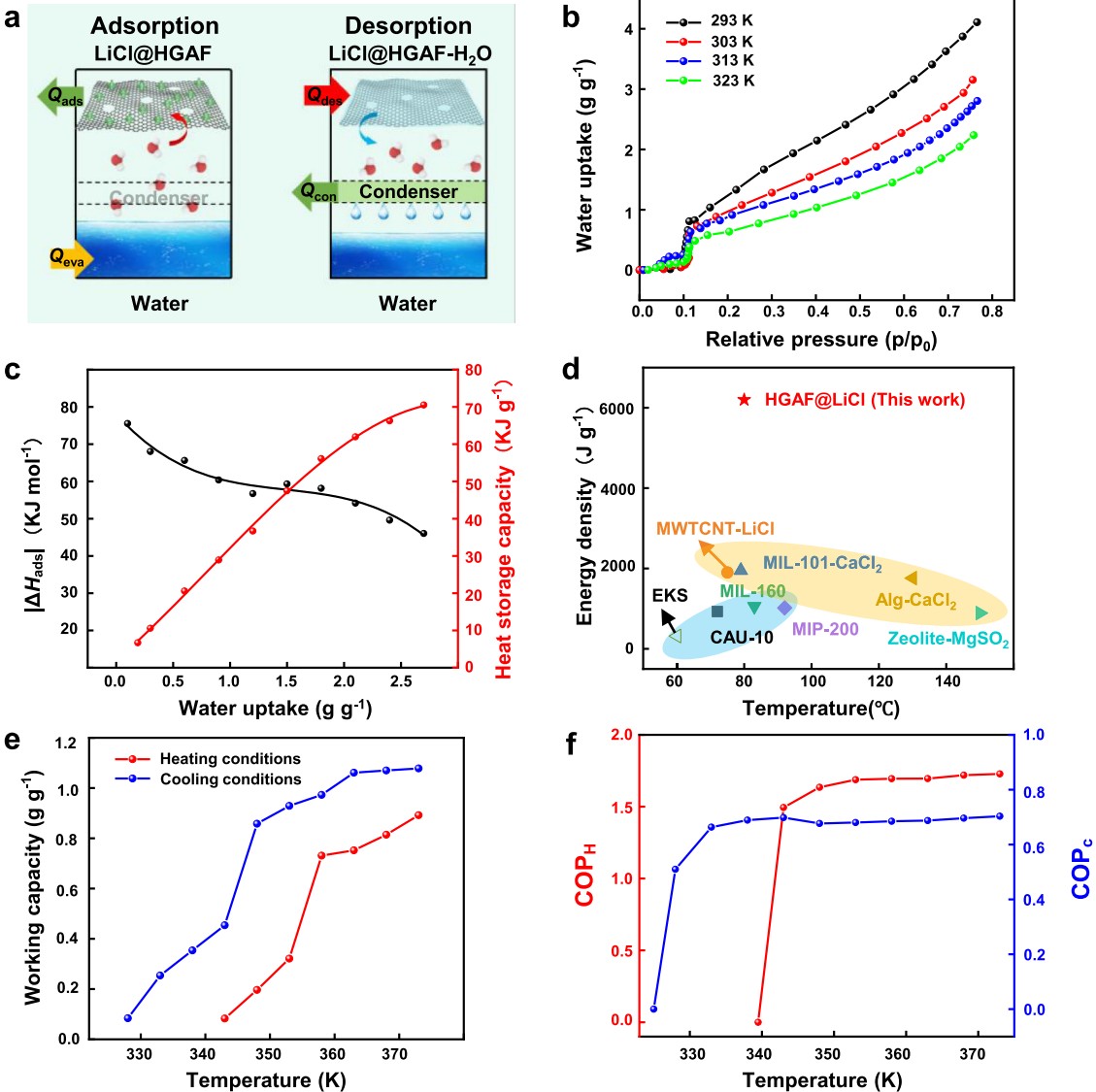

**Fig. 4 Heat allocation of LiCl@HGAFs. a** Working principle of LiCl@HGAF in an ATH device. During the adsorption process, the working fluid water absorbs heat ($Q_{eva}$) and evaporates, which is then captured by LiCl@HGAFs to release heat ($Q_{ads}$). And in the desorption process, the fibers absorb heat ($Q_{des}$) to release water vapor, which then condenses in the condenser and releases heat ($Q_{con}$). **b** Water sorption isotherms of LiCl@HGAFs-7 at 293 K, 303 K, 313 K, and 323 K. Source data are provided as a Source Data file. **c** Isosteric enthalpy of adsorption for water at LiCl@HGAFs-7 (black) and the corresponding heat storage capacity (red). Source data are provided as a Source Data file. **d** Comparison of energy density among our LiCl@HGAF sorbent, the reported MOFs, salt@MOF sorbents, and commercial hygroscopic fibers[35, 57–62]. Source data are provided as a Source Data file. **e** Plots of working capacity as a function of driving temperature for cooling conditions ($T_{eva} = 283$ K, $T_{ads} = 303$ K and $T_{con} = 303$ K) and heating conditions ($T_{eva} = 288$ K, $T_{ads} = 313$ K, and $T_{con} = 313$ K). Source data are provided as a Source Data file. **f** Calculation of the COP values for cooling and heating at different driving temperatures. Source data are provided as a Source Data file.

Fig. 27). Therefore, the working capacity of LiCl@HGAF at different temperatures can be obtained. At high temperatures, chemisorption plays a key role in water uptake, while the solution absorption improves water uptake at low temperatures. And the change of water vapor pressure does not significantly affect the capacity of chemical adsorption, but will increase the absorption capacity of the solution. Therefore, the LiCl@HGAF sorbent shows the flexibility of water uptake at different temperatures and water vapor pressures and becomes more adaptable than most reported adsorbents[32–34].

LiCl@HGAF exhibits high working capacity ($w = 0.2$ g g$^{-1}$) at $P/P_0 = 0.1$. It is worth noting that in the thermal application, it is more appropriate for working at low vapor pressure ($P/P_0 < 0.1$), which can reduce the use of compressors or increase the

evaporation temperature[35]. The heat storage capacity ($C_{HS}$) can be determined by:[36]

$$C_{HS} = \frac{\Delta H_{ads} \Delta w}{M_w} \qquad (2)$$

where, $M_w$ is the water molar weight and $\Delta H_{ads}$ is the isosteric enthalpy of sorption for water, which can be calculated using the Clausius-Clapeyron equation from two adsorption isotherm curves at different temperatures (details in Supplementary Section 1)[37].

It is found that as the water uptake increases, $|\Delta H_{ads}|$ decreases rapidly (from ~67 kJ mol$^{-1}$ to ~47 kJ mol$^{-1}$) (Fig. 4c), and that $|\Delta H_{ads}|$ is 63 kJ mol$^{-1}$ for a working capacity of 0.6 g g$^{-1}$. The initial high enthalpy of sorption corresponds to the formation of hydrates, where the water molecules are strongly bounded. The

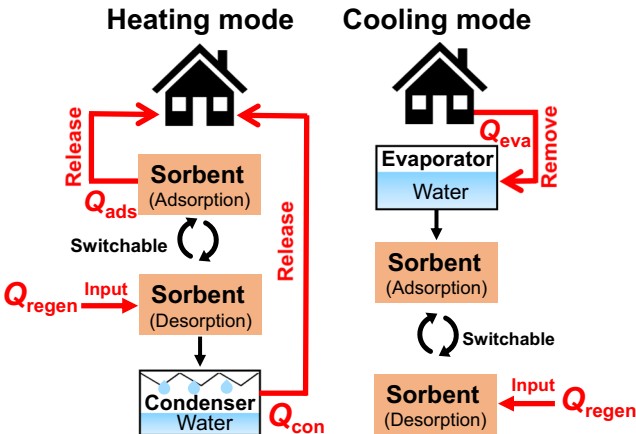

**Fig. 5 Heat transfer between the sorbent and the working fluid water for heating and cooling.** In the heating mode, the sorbents LiCl@HGAFs capture water molecules and release adsorption heat ($Q_{ads}$) to the house for indoor heating. As the adsorbent will become saturated with water, regeneration is required. Energy is taken up at a relatively high temperature ($Q_{regen}$) to desorb the water, which is subsequently condensed, releasing heat at an intermediate temperature ($Q_{con}$) to the house. Both of the released $Q_{ads}$ and $Q_{con}$ contribute to indoor heating. In the cooling mode, heat is taken up from the house by the evaporation of the working fluid ($Q_{eva}$), driven by the water sorption of the sorbents LiCl@HGAFs. Therefore, one can operate such a switchable sorption cycle as a heat pump to produce heating using $Q_{con}$ and $Q_{ads}$, or to produce cooling by using $Q_{eva}$.

sorption enthalpy of this process is the reaction enthalpy of the hydration, which value is usually higher than 60 kJ mol$^{-1}$ [32,35]. After the water sorption, the enthalpy performs a decrease to 47 kJ mol$^{-1}$, as close as the enthalpy of water evaporation (44 kJ mol$^{-1}$), due to the formation of an aqueous LiCl solution. The heat storage capacity ($C_{HS}$) for LiCl@HGAF-7 is calculated to be 6.93 kJ g$^{-1}$ (=0.19 kW h kg$^{-1}$), which is 1.68 times higher than that required by the U.S. Department of Energy (DOE) with the value of 0.071 kW h kg$^{-1}$ [36]. Furthermore, in comparison with other reported sorbents, our LiCl@HGAFs exhibit greater water uptake within a wide RH range (Fig. 3e) and it has the distinct advantages of high energy density for thermal storage applications (Fig. 4d).

The coefficient of performance (COP) is defined as useful energy output divided by the required energy as input, which is a commonly adopted indicator of the thermodynamic efficiency of the cycling process and depends strongly on the operating conditions (Fig. 5). The heating model (COP$_H$) and cooling model (COP$_C$) are given by Eqs. 3 and 4, respectively[38].

$$COP_H = \frac{-(Q_{con} + Q_{ads})}{Q_{regen}} \tag{3}$$

$$COP_C = \frac{Q_{eva}}{Q_{regen}} \tag{4}$$

The COP$_H$ values can range from 1 to 2. A high COP$_H$ value indicates the high energy efficiency in the heating mode. In this work, the COP$_H$ value obtained at different desorption temperatures is shown in Fig. 4f. The COP$_H$ is low at low temperature owning to the small working capacity, but it will increase abruptly as the desorption temperature rises (Fig. 4e). When the desorption temperature is 343 K, the COP value can reach close to 1.5. As for a heat pump, an evaporator temperature of 288 K, a heating or sorption temperature of 313 K, and a desorption temperature of 373 K were used. In these conditions, the COP$_H$

can reach 1.73 for LiCl@HGAF-7. In comparison, the highest COP$_H$ value (1.65) for salt@silica gel selective water sorbent has been reported in the literature, and the desorption temperature was higher in the range of 398–423 K, requiring more energy input[39]. For the sorbent-sorbate pairs such as MIL-101-methanol, SG/LiBr-methanol, CAU-3-ethanol, Maxsorb III-ethanol, and Ax-21-ammonia, the COP$_H$ only ranges from 1.0 to 1.2[40].

To evaluate the fiber as sorbents for adsorption chillers, we selected sorption temperature $T_{ad} = 303$ K and evaporation temperature $T_{evap} = 283$ K in the sorption-based cooling cycle, which is a typical value for practical applications[41]. The working capacity can be determined as 1.1 g g$^{-1}$ due to the complete desorption at $T_{des}$ (Fig. 4e). The COP$_C$ for LiCl@HGAFs-7 can reach 0.7 at the desorption temperature of 373 K. Aside from the COP$_C$ value, the specific cooling power (SCP) is another factor to demonstrate the efficiency of adsorption chillers. The average SCP is defined as the ratio of cooling power per mass of sorbent per cycle time, which describes the effectiveness of the system during the cooling process[25]. Finally, an SCP value of 297 W Kg$^{-1}$ was calculated (details in Supplementary Note 2), outperforming conventional adsorbents (e.g., silica gel: 63.4 W Kg$^{-1}$; activated carbon: 65 W Kg$^{-1}$; zeolite 13-X: 25.7 W Kg$^{-1}$)[42,43]. Our adsorbents demonstrated here encompass the optimal water-sorption properties with the high working capacities and specific energy capacities ever attained under adsorption-driven cooling and adsorption-driven heat pump working conditions

**Enhancement of microwave absorption performance based on moisture sorption.** Water has a frequency-dispersive permittivity in microwave frequencies and high transmittance characteristics, which can be seen as a promising candidate for designing broadband absorbers. Therefore, the fiber may obtain better microwave absorption performance after absorbing moisture. To achieve good absorption, the absorbing material must meet two criteria: (1) The incident electromagnetic wave can fully enter the interior of the material without reflection on the surface. That is, the matching characteristics of the material; (2) The electromagnetic wave entering the material can be quickly attenuated. To explore the microwave absorption performance, the relative complex permittivity (real part $\varepsilon'$ and imaginary part $\varepsilon''$) were measured in the frequency range of 1–18 GHz via the coaxial line method[44,45]. $\varepsilon'$ represents the dielectric and polarization property of material, and $\varepsilon''$ stands for the dielectric loss of materials. The $\varepsilon'$ values of LiCl@HGAFs and LiCl@HGAFs-H$_2$O tend to decrease with the increases in frequency, and the $\varepsilon''$ values for LiCl@HGAFs-H$_2$O reach their extremums at 6–8 GHz (Supplementary Fig. 29). The change of complex permittivity of all samples is closely related to the Debye relaxation process in the frequency range. With the effect of the applied electric field, the dipoles in the composites are deflected with the direction of the applied electric field, resulting in polarization loss. When the rearrangement of dipoles cannot keep its rhythm up with the rapidly changing external electromagnetic field, dielectric relaxation loss will occur[46]. The mechanism of microwave absorption is shown in Fig. 6. The conductivity of holey graphene oxide (HGO) benefits conductivity loss and layered sheet-like HGO could result in multiple microwave reflections and thus further enhance the electromagnetic damping capacity (Fig. 6a). The interface polarization is generated by charge aggregation on the interface between the graphene and LiCl (Fig. 6b), where the oxygen-containing groups and defects on the HGO introduce more polar centers (Fig. 6a–c), and the addition of water further increases the dielectric loss (Fig. 6c). To understand the microwave absorption mechanism of the fiber, Cole-Cole curves were plotted to examine the polarization relaxation behaviors according to Debye

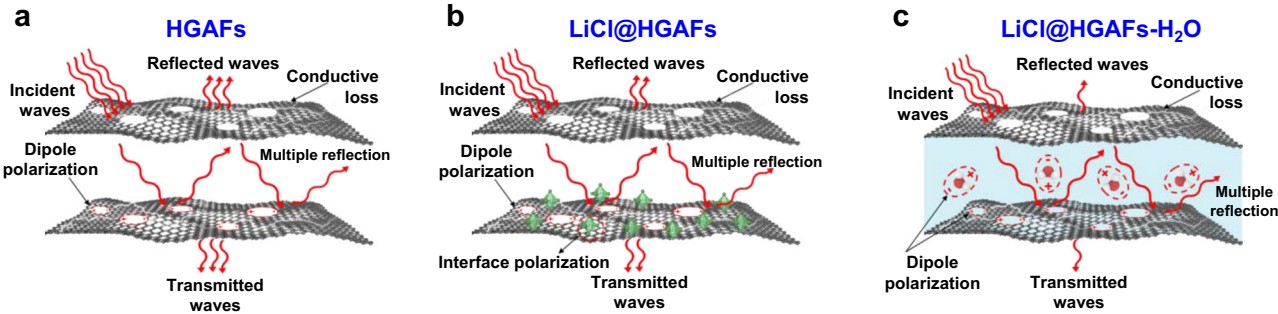

**Fig. 6 Schematic diagram of the absorption mechanism. a** HGAFs, **b** LiCl@HGAFs, and **c** LiCl@HGAFs-$H_2O$. The blue-shaded region in panel **c** refers to the liquid water in the fibers. The absorption mechanism mainly includes conductive loss, interface polarization, and dipole polarization. The incident wave will have multiple reflections in the material, and the above loss process will occur continuously. The dipole polarization process is greatly enhanced when water is present in the material.

**Fig. 7 Microwave absorption characterization.** Reflection loss (RL) value of **a** HGAFs, **b** LiCl@HGAFs, and **c** LiCl@HGAFs-$H_2O$. Source data are provided as a Source Data file. Calculated $|Z_{in}/Z_0|$ values of **d** HGAFs, **e** LiCl@HGAFs, and **f** LiCl@HGAFs-$H_2O$ at different thicknesses. Source data are provided as a Source Data file. **g** Cole-cole plot of LiCl@HGAFs-$H_2O$. Source data are provided as a Source Data file. **h** The $|Z_{in}/Z_0|$ value of HGAFs, LiCl@HGAFs, and LiCl@HGAFs-$H_2O$ with a thickness of 2.5 mm. Source data are provided as a Source Data file. **i** The integrated effective absorption bandwidth (EAB) at the scope of 1–18 GHz. The S, C, X, and Ku in panel (**i**) refer to different microwave bands, 2–4 GHz, 4–8 GHz, 8–12 GHz, and 12–18 GHz, respectively. Source data are provided as a Source Data file.

relaxation theory (details in Supplementary Note 5)[47]. LiCl@H-GAFs-$H_2O$ display an undulant curve containing many semicircles and a linear tail representing the multiple polarization relaxations, such as dipolar polarization, interfacial polarization, and electron conduction loss when electromagnetic waves pass through the LiCl@HGAFs-$H_2O$ (Fig. 7g). Moreover, water can interact with wide microwave frequencies and thus exhibit wide microwave absorption. (Fig. 6c)[48].

According to Supplementary Note 4 and the transmission line theory, the electromagnetic wave absorption properties (reflection loss, RL) of different HGAF profiles were calculated (Fig. 7a–c and Supplementary Fig. 30). Generally, RL ≤ −10 dB means more than 90% absorption of electromagnetic waves[49]. Compared with the HGAF, the microwave absorption performance of LiCl@HGAF is significantly improved by the enhanced polarization in the HGO network (Figs. 6b and 7b). After filling of $H_2O$, LiCl@HGAF-$H_2O$ exhibits a wide-band microwave absorption ranging from 8.31 GHz to 18 GHz at the sample thickness of 2.5 mm and the minimum reflection loss can reach −27.9 dB at 17.3 GHz (Fig. 7c). Effective absorption bandwidth (EAB) is an important property to evaluate the wideband characteristic of microwave absorbing materials. The acceptable RL value for EAB is −10.0 dB, which means that the incidence wave is attenuated by 90%. The integrated EABs of LiCl@HGAF-$H_2O$ along with some conventional homologous 2D graphene-based microwave absorption materials are displayed in Fig. 7h. In comparison, LiCl@HGAF-$H_2O$ presents an excellent wideband EM absorption ability with almost integrated EAB covering all of Ku band, X band, C band, and even part of S band. Compared with LiCl@HGAF, LiCl@HGAF-$H_2O$ has unique advantages in broadband microwave absorption. However, as the water content increases, the reflection loss of the samples (2.5 mm) increases first and then decreases (Supplementary Fig. 30g). This is due to the excessive water content makes the impedance matching worse (Supplementary Fig. 31b–e). Impedance matching is a crucial factor for microwave absorption performance, which describes the ability of electromagnetic wave attenuation by generating more input impedance rather than reflection to the air. The impedance matching ($Z = |Z_{in}/Z_0|$, Fig. 7d–f) between materials and space is calculated by Supplementary Note 4. When the value of $|Z_{in}/Z_0|$ is close to 1 (the impedance of free space), the sample would exhibit good impedance match performance, which is a prerequisite for excellent microwave absorption. As shown in Fig. 7h and Supplementary Fig. 25, both HGAF, LiCl@HGAF show relatively poor impedance matching areas in the range of 0.8–1.2. However, the impedance matching area of LiCl@HGAF-$H_2O$ in this range increases markedly, revealing that the impedance matching has been obtained successfully with the confinement of water (Fig. 6c). To obtain high microwave absorption performance, the material needs to not only meet impedance match, but also a large attenuation constant (α) is required to satisfy the large energy loss[50]. The attenuation constant α (details in Supplementary Section 2) determines the attenuation capability of the input electromagnetic waves[51]. Here, the α values decrease with improving the water content in the LiCl@HGAF-7 (Supplementary Fig. 32). Although the sample with a water content of 2 g g$^{-1}$ exhibits a moderate α, the impedance matching allows EM waves to enter the absorbing materials as much as possible, which is the prerequisite to obtain a better EM wave absorbing property[52]. Since the microwave absorption stability of hygroscopic holey graphene aerogel fibers containing water is crucial in real applications, we characterized the long-time stability of LiCl@HGAF-$H_2O$ fibers in an electromagnetic environment with the power of +10 dBm. The samples with variant thicknesses exhibit stable microwave sorption behavior up to 12 h (Supplementary Fig. 33a–f). The maximum absorptivity of the fibers was above 99% invariably (Supplementary Fig. 33g).

## Discussion

In summary, we report a strategy to develop LiCl@HGAFs with superior moisture capture, heat allocation, and microwave absorption functionalities. The hygroscopic nature of LiCl, combined with the high surface area and interconnected porous network of graphene host fiber, ensure that LiCl@HGAF obtains ultra-high water sorption capacity and uptake kinetics. Besides, due to the outstanding photo-thermal and electro-thermal effects, the LiCl@HGAFs can realize the regeneration in multiple pathways and with low energy input. On the other hand, the LiCl@HGAFs are demonstrated in the adsorption-based heat transfer devices with high heat storage capacity and desirable coefficient of performance both in the heating mode and cooling mode. The incorporation of efficient "adsorbent-water" in our AHT devices is an attractive alternative solution to develop a green and sustainable technology to meet the surge in global energy demands for heating and cooling. Furthermore, when containing saline water in the porous confinements, the LiCl@HGAFs can effectively absorb the electromagnetic wave in a wide range of bandwidth, with outstanding impedance matching and a large attenuation coefficient. In short, the holey graphene aerogel fibers combining with hygroscopic LiCl salt introduced here may offer important alternatives for developing multifunctional materials for water harvest, thermal energy utilization, and microwave adsorption, as well as open unexplored opportunities for aerogel fiber-related technology in various applications. It is envisioned that our results will also spur future efforts for the development of advanced adsorbents, dehumidifiers, sorption-based heat transfer systems, adsorption-driven refrigeration, and beyond.

## Methods

**General**. The experimental materials and detailed calculations for heat allocation and microwave absorption applications are given in the Supplementary Methods.

**Synthesis of GO**. GO aqueous dispersion was prepared from natural graphite powder according to a modified Hummers method[53]. Briefly, 12 g graphite powder, 100 ml $H_2SO_4$, 10 g $K_2S_2O_8$, and 10 g $P_2O_5$ were added to a 250 ml flask and the mixture was kept at 80 °C for 4.5 h. After cooling to room temperature, the mixture was diluted with 1 L water and vacuum-filtered and washed with water using a 0.22-μm pore polycarbonate membrane. After drying, the 4 g solid was added into 160 ml concentrated $H_2SO_4$ (0 °C), and then 20 g $KMnO_4$ was added slowly under continuous stirring. After the introduction of $KMnO_4$, the mixture was heated to 35 °C and stirred for 2 h. The mixture was then diluted with 1.2 L water, followed by dropwise addition of 10 ml 30% $H_2O_2$. Using the centrifugation washing method, the precipitate was repeatedly washed with water, 1 M HCl solution, and water successively. Finally, we obtained GO aqueous dispersions after ultra-sonication for 2 h.

**Synthesis of holey graphene oxide-LC**. The holey graphene oxide was synthesized according to our previous work[20]. Briefly, 50 ml $H_2O_2$ (30% aqueous solution) was added into 500 ml GO aqueous dispersion (2 mg/ml) and stirred for 10 min to obtain a uniform mixture. And the mixture was then heated at 100 °C under stirring for different periods. The reaction time of the mixture was 0.5, 1, 1.5, 2 h, respectively, denoted as HGO-1, HGO-2, HGO-3, and HGO-4. Then the mixture was centrifuged and washed with water 3 times to remove the impurities. Finally, the obtained HGO aqueous dispersion was concentrated by centrifuging at a high speed for 4 h to obtain HGO-LC.

**Preparation of holey graphene aerogel fiber**. The holey graphene oxide liquid crystal (25 mg/ml) was spinning into 0.5 wt% $CaCl_2$ aqueous solution and the obtained holey graphene oxide hydrogel fiber was immersed into 10% hydroiodic acid aqueous solution at 60 °C for 5 h. Followed by washing at least four times with absolute ethyl alcohol to replace the water and supercritical drying with $CO_2$ (40 °C, 10 MPa) for 12 h, the HGAF was obtained.

**Aging effect of HGAF**. The as-prepared holey graphene oxide hydrogel fibers were aged for 1, 2, and 3 days at room temperature. Then the fibers were immersed into 10% hydroiodic acid aqueous solution at 60 °C for 5 h. After that, they were washed at least four times with absolute ethyl alcohol to replace the water followed by supercritical drying with $CO_2$ (40 °C, 10 MPa) for 12 h. Then, the specific surface area, electrical conductivity, tensile strength, and average diameter of the aged fibers were characterized.

**Preparation of LiCl@holey graphene aerogel fiber**. The impregnation method was applied to coat LiCl into holy graphene aero fibers. The periods of LiCl solutions (concentration of 3%, 5%, 7%, respectively) were prepared by adding LiCl

particles to water and stirring for 30 min. The samples were LiCl@HGAF-3, LiCl@HGAF-5, and LiCl@HGAF-7. Subsequently, the holey graphene aerogel fiber was impregnated to the solution for 24 h. Due to the hydrophobicity of holey graphene aerogel fiber, the fiber was difficult to submerge in LiCl solution. Therefore, we infiltrated the fiber with ethanol before the impregnating process. Finally, the freeze-drying process was carried out.

**Characterizations**. The morphologies and structures of holey graphene aerogel fibers and LiCl@holey graphene aerogel fibers were characterized by a scanning electron microscope (S-4800) operated at 15 kV and a transmission electron microscope (Tecnai G2 F20 S-Twin). $N_2$ gas adsorption isotherms were measured with a physisorption apparatus (ASAP 2020, Micromeritics Instrument) at 77 K. Raman spectra were recorded on a LabRam HR Raman spectrometer with 50 W He-Ne laser operating at 632 nm with a CCD deter. X-ray diffraction (XRD) patterns were recorded on a D8 Advanced spectrometer with an angular range of 10–90° (2 thetas). Thermal gravimetric analysis and DTG were carried out using a TG 209F1 Libra (NETZSCH) analyzer with a heating rate of 10 °C min$^{-1}$ in a nitrogen atmosphere. DSC analysis was performed on a DSC 200F3 NETZSCH with a heating and cooling rate of 10 °C min$^{-1}$. The electric resistances of the aerogel and its composites were measured by using a CHIChief 600D electro-chemical workstation, and the electric conductivity can be calculated by the equation: $\kappa = IL/US$, where $\kappa$ is the electric conductivity, $I$ is the current that crosses the sample, $U$ is the voltage applied in the sample, $L$ is the length of sample current goes through, and $S$ is the cross area of current. The stress-strain curves were measured by an Instron 3365 tensile testing machine and the bending stiffness can be calculated by the equation: $R_f = \pi ED^4/64$, where $E$ is the elastic modulus, $D$ is the average diameter of the fiber. The contact angle was measured by OCA 15EC DataPhysics Instruments GmbH. Infrared photos were taken with a MinIR (M1100150) camera. The XPS spectra were measured by Escalab 250Xi, Thermo Scientific. The temperature-time curves were measured and recorded by the thermal couple and Keysight 34970 Data Acquisition. Water vapor adsorption isotherms were measured by a volumetric method using a volume method vapor sorption analyzer (BSD-PMV2) at the temperature of 293 K, 303 K, 313 K, and 323 K. Before the measurement, the sample was pre-activated at 100 °C for 6 h to remove all residual water. The sorption kinetics curves and cyclic capacity were measured in a glove box with a balance measuring the weight of the fibers under a constant temperature and humidity. The cycle test was performed at 25 °C, 90RH% for 6 h for adsorption, and at 10RH%, 60 °C for 60 min for desorption.

**Dehumidification experiment of different moisture sorption materials**. LiCl@HGAFs, as well as other moisture sorption materials including color-changing silica gel (methyl violet@silical gel), commercial hygroscopic fibers (EKS fibers from TOYOBO CO., LTD), LiCl, UiO-66 (zirconium 1,4-dicarboxybenzene MOF), LiCl@UiO-66, active carbon fiber loaded with lithium chloride (LiCl@ACF), were conducted with the dehumidification experiment. A series of moisture sorption materials in two sets of experiments were prepared for the dehumidification test. One group of them were in the same mass (2 g), and the other group of them were in the same packing volume (5 cm$^3$). At room temperature, the sorption materials were placed in a sealed chamber with a size of 60*50*50 cm and an initial relative humidity of 90%, and the relative humidity in the chamber was detected after 6 h.

**Microwave sorption stability of LiCl@HGAF-H$_2$O**. We placed the LiCl@HGAF-H$_2$O in the testing chamber and set the test power to +10 dBm (10 mW) to keep the materials in the electromagnetic environment all the time. After that, this test was conducted every 2 h to obtain the changes in the microwave absorption properties of the materials.

## Data availability

All data generated in this study are provided in the Supplementary Information/Source Data file or from the corresponding author upon request. Source data are provided with this paper.

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

## Acknowledgements

This work was supported by the National Key Research and Development Program of China (no. 2020YFB1505703) to Z.S., the Royal Society Newton Advanced Fellowship (no. NA170184) to X.Z., the National Science Fund for Distinguished Young Scholars (no. 52025034) to J.K., the National Natural Science Foundation of China (no. 52173052) to X.Z., the National Science Foundation of Jiangsu Province (no. BK20211099) to Z.S., and the Youth Innovation Promotion Association of Chinese Academy of Sciences (no. 2022325) to Z.S. We appreciate the beneficial discussion of theoretical data fitting with Dr. Mengchuang Zhang and Zhizhuo Zhang from Northwestern Polytechnical University.

## Author contributions

X.Z. supervised the project. Y.H., Z.S., and J.K. conceived and designed the experiments. Y.H. performed experiments and acquired data. Y.H., Z.S., J.K., and X.Z. analyzed the data and drafted the manuscript. All authors supported revising the manuscript.

## Competing interests

The authors declare no competing interests.
