## [Peer Review File · Nature Communications]

REVIEWER COMMENTS

Reviewer #1 (Remarks to the Author):

This paper reports fabrication and performance of hygroscopic holey graphene aerogel fibers which can be applied for moisture capture, heat allocation, and microwave absorption. This theme is interesting and it is worthy to be studied. The paper has clearly presented the detailed fabrication process and performance data of LiCl@HGAFs. This paper should be published. Some comments here can help to improve this paper.

1. In Fig. 2f, the desorption temperature of LiCl@HGAFs is very low when compared to other LiCl composite adsorbents. Can the authors explain it?
2. The small picture in Fig. 3f does not distinguish the difference of heating and cooling modes. Please correct it.
3. On the right side of Eq. S5, it misses the sensible heat items.
4. In Eq. S6, T_{con} and T_{cond} are not correct. They should be T_{abs} .
5. $Q_{sorption}$ is redundant and Eq. S7 is not correct.
6. ΔH_{vap} is redundant. ΔH_{vap} should be replaced by ΔH_{eva} and M_w should be deleted in Eq. S8.

Reviewer #2 (Remarks to the Author):

The authors have reported a practical strategy to fabricate hygroscopic holey graphene aerogel fibers with integrated functions of highly efficient moisture capture, heat allocation, and microwave absorption. The fabricated hygroscopic holey graphene aerogel fibers named LiCl@HGAFs realize the water sorption capacity over 4.15 g g⁻¹ at $p/p_0=0.9$, surpassing conventional moisture adsorbents due to the super high surface area and high water uptake kinetics for water transport. Furthermore, the studied sorbents can be regenerated using both photothermal and electrothermal methods with little energy input. The work is novel, and it is very exciting. This reviewer suggests the authors to revise their manuscript considering the following comments.

The aging effect of the fabricated holey graphene aerogel fibers should be given. Provide cyclic test results of water uptake onto LiCl@GAFs and LiCl@HGAFs.

Provide TEM images of LiCl@GAFs and LiCl@HGAFs after cyclic test and compare those with the TEM images of the synthesized LiCl@GAFs and LiCl@HGAFs before any experiment.

Fit the water uptake data with Freundlich and S-B-K models.

Consult the following papers:

R.H. Mohammed et al., Revisiting the adsorption equilibrium equations of silica-gel/water for adsorption cooling applications, *International Journal of Refrigeration* 86 (2018) 40–47.

B.B. Saha et al., Computer simulation of a silica gel water adsorption refrigeration cycle - the influence of operating conditions on cooling output and COP, *ASHRAE Transactions*, Vol. 101, No. 2, pp. 348-357, 1995.

This work reports hygroscopic holey graphene aerogel fibers for highly efficient moisture capture, heat allocation and microwave sorption. A series of experiments were conducted to support the conclusions and claims. However, there is lack of novelty and there are still a few questions to be resolved:

1. For moisture sorption, sorption capacity defined as weight of sorbed water/weight of sorbent was adopted to depict the water sorption capacity. Compared with other sorbents, the author claimed that the holey graphene aerogel have the highest sorption capacity. However, the excellent sorption capacity of the fiber is mainly due to its low density, which is not available in real applications. I suggest that the author supplement the following experiments: various moisture sorbents with the same weight or volume are put into a closed space with the same initial relative humidity; after a period of time, to evaluate which sorbent reduces the relative humidity most.
2. For microwave sorption, this work show that the fiber possess good microwave sorption own to the entrapped water. Therefore, an experiment about the sorption stability of microwave should be supplemented in that the water will be heated after microwave sorption, which will inevitably lead to the decrease of sorption performance.

Reply to reviewers

For Reviewer #1:

We appreciate the reviewer's positive evaluation of our manuscript: "*This paper reports fabrication and performance of hygroscopic holey graphene aerogel fibers which can be applied for moisture capture, heat allocation, and microwave absorption. This theme is interesting and it is worthy to be studied. The paper has clearly presented the detailed fabrication process and performance data of LiCl@HGAFs. This paper should be published.*"

We thank the reviewer very much for the constructive comments and suggestions to improve our manuscript greatly. We have addressed all the issues in a point-by-point manner and all the changes to the original manuscript are highlighted in blue.

Comment 1: "In Fig. 2f, the desorption temperature of LiCl@HGAFs is very low when compared to other LiCl composite adsorbents. Can the authors explain it?"

Response: Fig. 2f refers to the photo-response of HGAFs and LiCl@HGAFs under one-sun irradiation. The highest temperature reaches around 50 °C for HGAFs and 47 °C for LiCl@HGAFs, respectively. And partial desorption of LiCl@HGAFs occurs under the photo-thermal condition.

As the reverse of the adsorption process (Figure R1), there are three steps successively for fully desorption¹:

- 1) Water evaporation from LiCl solution
- 2) The crystallization of LiCl·H₂O
- 3) The chemical desorption of LiCl·H₂O to LiCl

The complete desorption temperature is determined by chemical desorption of LiCl·H₂O, the equilibrium can be expressed as following equation:¹

$$\ln \frac{p}{p_0} = -\frac{8868.18}{T + 273.15} - 4 \ln(T + 273.15) + 44.589$$

It can be seen from the above equation that the desorption temperature of the LiCl@HGAFs depends on the water vapor pressure. In our operating water vapor pressure (1.2 kPa) for regeneration, the theoretical desorption temperature of LiCl·H₂O is 69 °C . Since the highest temperature of is LiCl@HGAFs is 47°C under one-sun irradiation, the LiCl solution evaporates to form LiCl·H₂O while dehydration reaction of LiCl·H₂O does not occur according to the phase balance between LiCl and water². Therefore, LiCl@HGAFs process an incomplete water desorption (83.4 %) by processing evaporation of LiCl solution to LiCl·H₂O under the photo-thermal condition. On the other hand, under the electro-thermal condition (10 V), since the temperature of LiCl@HGAFs

(81.6 °C) is higher than the dehydration temperature of LiCl·H₂O (69 °C), the LiCl@HGAFs can release all captured water (with the final form of LiCl).

Figure R1. Water sorption isotherms of LiCl@HGAFs at 313 K.

Some LiCl composite sorbents are listed in Table R1. Compared with other LiCl composite sorbents, LiCl@HGAFs show a similar desorption temperature for both complete desorption and partial desorption. The difference in the desorption temperature is attributed to the difference in both thermal conductivity and water transport pathways of porous matrix materials to confine the LiCl. It can be seen that complete desorption usually requires a higher desorption temperature, while partial desorption

requires a lower temperature, and the degree of desorption largely depends on the temperature and relative pressure during desorption⁵⁻⁷. For example, PNIPAM-Silica gel-LiCl can be partially desorbed at 40°C and 7.38 kPa (10 RH%), but the degree of desorption is only 56% and when the temperature varied from 40 to 50 °C, the degree of desorption increased by 24%⁷.

Table R1. Desorption temperature of a series LiCl composite sorbents.

Desorption extent	LiCl composite adsorbent	Operating desorption temperature	Ref.
Complete desorption	LiCl@rGO-SA	85°C	1
	LiCl@MIL-101(Cr)	83°C	2
	HCS-LiCl	80°C	3
	PAN/MIL@LiCl NFM	100°C	4
	LiCl@HGAFs	81.6°C	This work
Partial desorption	PVA-LiCl	52°C (desorption degree 88%)	5
	LiCl-Coated BCS	51.3°C (desorption degree ~90%)	6
	PNIPAM-Silica gel-LiCl	40°C (desorption degree 56%)	7
	LiCl@HGAFs	47°C (desorption degree 83.4%)	This work

1. Xu, J. et al., Ultrahigh solar-driven atmospheric water production enabled by scalable rapid-cycling water harvester with vertically aligned nanocomposite sorbent. *Energy Environ. Sci.*, 2021,14, 5979-5994.

2. Xu, J. X. et al., Efficient solar-driven water harvesting from arid air with metal organic frameworks modified by hygroscopic salt. *Angew. Chem., Int. Ed.*, 2020, 59, 5202–5210.
3. Li, R. et al., Improving atmospheric water production yield: Enabling multiple water harvesting cycles with nano sorbent. *Nano Energy*, 2020, 67, 104255.
4. Zhang, Y. et al., Super hygroscopic nanofibrous membrane-based moisture pump for solar-driven indoor dehumidification. *Nat. Commun.* 2020, 11, 3302.
5. Dai, L. et al., Sorption and regeneration performance of novel solid desiccant based on PVA-LiCl electrospun nanofibrous membrane. *Polym. Test.* 2017, 64, 242-249.
6. Gong, F. et al., Agricultural waste-derived moisture-absorber for all-weather atmospheric water collection and electricity generation. *Nano Energy*, 2020, 74, 104922.
7. Ma, Q. et al., Preparation and characterization of thermo-responsive composite for adsorption-based dehumidification and water harvesting. *Chem. Eng. J.*, 2022, 429, 132498.

Revision:

The following content has been added in the manuscript on page 13:

LiCl@HGAFs possess both high solar-thermal conversion and high electro-thermal capability...The highest temperature reaches around 50°C for HGAFs and 47°C for LiCl@HGAFs, respectively. The desorption of LiCl@HGAFs was carried out under a typical water vapor pressure of 1.2 kPa. Considering that the desorption temperature of LiCl@HGAFs depends on the dehydration of LiCl·H₂O, the theoretical desorption temperature to the product of LiCl is 69°C according to Clausius–Clapeyron equilibrium equation³⁰. Therefore, under the photo-thermal

condition (47°C), LiCl@HGAFs undergo the desorption from LiCl solution to LiCl·H₂O, where the regeneration degree can reach 83.4% (Supplementary Fig. 20)...

Comment 2: “The small picture in Fig. 3f does not distinguish the difference of heating and cooling modes. Please correct it.”

Response: We thank the reviewer for the comments on the improvement of the figures. In the heating mode, the sorbents LiCl@HGAFs capture water molecules and release adsorption heat for indoor heating (Figure R3, left). In the cooling mode, the evaporation of working fluid (water), driven by the water sorption of LiCl@HGAFs, enables indoor cooling by removing the heat of evaporation (Figure R3, right). The coefficient of performance (COP) is defined as useful energy output divided by the required energy as input, which is described as COP_H and COP_C for the heating mode and the cooling mode, respectively.

$$COP_H = \frac{-(Q_{con} + Q_{ads})}{Q_{regen}} \quad (3)$$

$$COP_C = \frac{Q_{eva}}{Q_{regen}} \quad (4)$$

Figure R2. Working principle of the heating mode and cooling mode. In the heating mode, the sorbents LiCl@HGAFs capture water molecules and release adsorption heat (Q_{ads}) to the house for indoor heating. As the adsorbent will become saturated with water, regeneration is required. Energy is taken up at a relatively high temperature (Q_{regen}) to desorb the water, which is subsequently condensed, releasing heat at an intermediate temperature (Q_{con}) to the house. Both of the released Q_{ads} and Q_{con} contribute to indoor heating. In the cooling mode, heat is taken up from the house by the evaporation of the working fluid (Q_{eva}), driven by the water sorption of the sorbents LiCl@HGAFs. Therefore, one can operate such a switchable sorption cycle as a heat pump to produce heating energy using Q_{con} and Q_{ads} , or to produce cooling energy by using Q_{eva} .

Revision: To distinguish the difference between heating and cooling modes, we have reorganized the inset in **Figure 3f** by showing the heat transfer between the adsorbent and the working fluid (as shown in revised Fig. 3).

Fig. 3 Heat allocation of LiCl@HGAFs. **a** Working principle of LiCl@HGAF in an ATH device. **b** Water sorption isotherms of LiCl@HGAFs-7 at 293 K, 303 K, 313 K, and 323 K. **c** Isothermic enthalpy of adsorption for water at LiCl@HGAFs-7 (black) and the corresponding heat storage capacity (red)...**f**. Calculation of the COP values for cooling and heating at different driving temperatures. The inset shows the heat transfer between the sorbent and the working fluid water in the heating and cooling mode. In the heating mode, the sorbents LiCl@HGAFs capture water molecules and release adsorption heat (Q_{ads}) to the house for indoor heating. As the adsorbent will become saturated with water, regeneration is required. Energy is taken up at a

relatively high temperature (Q_{regen}) to desorb the water, which is subsequently condensed, releasing heat at an intermediate temperature (Q_{con}) to the house. Both of the released Q_{ads} and Q_{con} contribute to indoor heating. In the cooling mode, heat is taken up from the house by the evaporation of the working fluid (Q_{eva}), driven by the water sorption of the sorbents LiCl@HGAFs. Therefore, one can operate such a switchable sorption cycle as a heat pump to produce heating using Q_{con} and Q_{ads} , or to produce cooling by using Q_{eva} .

Comment 3: “On the right side of Eq. S5, it misses the sensible heat items.”

Reply: We have carefully checked our calculation, and in the original manuscript, we considered the sensible heat term during the calculation but missed the writing when typing the eq. S5. Therefore, we have added the sensible heat items in the revised manuscript.

Revision:

Q_{eva} is the energy taken up by evaporation, Q_{cond} is the energy released by the condenser, Q_{ads} is the energy gained during the adsorption process, and Q_{regen} is the energy required by regeneration, they can be calculated by ²⁻⁴

$$Q_{eva} = \Delta H_{eva}(T_{eva})\Delta w \quad (\text{S3})$$

$$Q_{cond} = \Delta H_{cond}(T_{cond})\Delta w \quad (\text{S4})$$

Where ΔH_{eva} is evaporation enthalpy and ΔH_{cond} is the condensation enthalpy. T_{eva} and T_{con} are temperatures of evaporator and condenser, respectively.

$$\begin{aligned}
 Q_{ads} = & \int_{T_{des}}^{T_{IC}} C_p^{sorbent}(T) dT + \int_{T_{des}}^{T_{IC}} \rho_{liq}^{wf} w_{max} C_p^{wf}(T) dT \\
 & + \int_{T_{IC}}^{T_{con}} C_p^{sorbent}(T) dT + \int_{T_{IC}}^{T_{con}} \rho_{liq}^{wf} \frac{(w_{max} + w_{min})}{2} C_p^{wf}(T) dT \\
 & + Q_{sorption}
 \end{aligned}$$

(S5)

In the above equation, M_w is the molar mass of working fluid (water), ΔH_{ads} is the adsorption enthalpy, w_{max} and w_{min} are the maximal and minimum working capacity at sorption condition and desorption condition. T_{des} is the desorption temperature. T_{IC} is the terminal temperature of the sorbent after isosteric cooling. $C_p^{sorbent}$ and C_p^{wf} are the specific heat capacity of sorbent (Supplementary Figure 29) and working fluid. ρ_{liq}^{wf} is the density of working fluid. $Q_{sorption}$ is the energy released during adsorption of the sorbent, which can be calculated by:

$$Q_{sorption} = \frac{1}{M_w} \int_{w_{min}}^{w_{max}} \Delta H_{ads}(w) dw$$

Comment 4: “In Eq. S6, T_{con} and T_{cond} are not correct. They should be T_{abs} .”

Reply: We appreciate the reviewer for raising this point in Eq. S6. We have replaced T_{con} and T_{cond} with T_{abs} in Eq. S6. During the calculation, the values of T_{abs} and T_{con} are the same, so the calculation result is not affected.

Revision: Based on the reviewer's suggestion, we have revised Eq. S6 as follows:

$$Q_{regen} = \int_{T_{abs}}^{T_{des}} C_p^{sorbent}(T) dT + \int_{T_{abs}}^{T_{des}} \frac{w_{max} + w_{min}}{2} C_p^{wf}(T) - Q_{sorption} \quad (S6)$$

Here, T_{des} and T_{abs} refers to the desorption temperature of sorbent and the temperature of absorption.

Comment 5: " $Q_{sorption}$ is redundant and Eq. S7 is not correct."

Reply: The original Eq. S7 has been included in the revised Eq. S5 and therefore we delete the original Eq. S7.

Revision: The item $Q_{sorption}$ mentioned in Eq. S7 has been deleted. The order of new equations has been updated.

Comment 6: " ΔH_{vap} is redundant. ΔH_{vap} should be replaced by ΔH_{eva} and M_w should be deleted in Eq. S8"

Reply: In fact, the unit of ΔH_{vap} used in Eq. S8 is in KJ/mol, so M_w (Kg/mol) needs to be added. Based on the reviewer's comment, we have replaced ΔH_{vap} and deleted M_w in Eq. S8 to obtain the SCP with the unit of KJ/Kg.

Revision: Based on the reviewer's suggestion, we have revised the original Eq. S8 as follows (revised Eq. S7).

The average specific cooling power (SCP) can be calculated by the following equation:

$$SCP = \frac{0.8\Delta H_{eva} \Delta w}{\tau_{0.8ads} + \tau_{0.8des}} \quad (S7)$$

where ΔH_{vap} (KJ/Kg) is the water enthalpy of evaporation, Δw is the working capacity of the LiCl@HGAFs-7/H₂O pair, $\tau_{0.8ads}$ and $\tau_{0.8des}$ are the adsorption and desorption times with the conversion $q = 0.8$ (Supplementary Figure 18, 20), which are 5368 s and 1726 s, respectively.

We appreciate the reviewer for the constructive comments and suggestions to improve our manuscript, thanks again !

For Reviewer #2:

We appreciate the reviewer's positive evaluation of our manuscript: "*The authors have reported a practical strategy to fabricate hygroscopic holey graphene aerogel fibers with integrated functions of highly efficient moisture capture, heat allocation, and microwave absorption. The fabricated hygroscopic holey graphene aerogel fibers named LiCl@HGAFs realize the water sorption capacity over 4.15 g g⁻¹ at p/p₀=0.9, surpassing conventional moisture adsorbents due to the super high surface area and high water uptake kinetics for water transport. Furthermore, the studied sorbents can be regenerated using both photothermal and electrothermal methods with little energy input. The work is novel, and it is very exciting.*" We thank the reviewer a lot for the constructive comments and suggestions to improve our manuscript greatly. We have addressed all the comments in a point-by-point manner and all the changes to the original manuscript are highlighted in blue.

Comment 1: "The aging effect of the fabricated holey graphene aerogel fibers should be given. Provide cyclic test results of water uptake onto LiCl@GAFs and LiCl@HGAFs."

Reply: We appreciate the reviewers for this suggestion. The influence of the aging effect is important for aerogel materials. We have prepared holey graphene oxide hydrogel fibers and aged them for 1, 2, and 3 days

at room temperature. Then the fibers were immersed into 10% hydroiodic acid aqueous solution at 60°C for 5 h. After that, they were washed at least four times with absolute ethyl alcohol to replace the water followed by supercritical drying with CO₂ (40°C, 10 MPa) for 12 h. And then we tested the specific surface area, electrical conductivity, tensile strength, and average diameter of the aged fibers. In the original manuscript, we have provided the cyclic test results of water uptake onto LiCl@HAGFs in Figure 3f. According to the reviewer's suggestion, we have also provided cyclic test results of water uptake onto LiCl@GAFs (Supplementary Figure 22).

Revision: The following sentences were added on pages 8-9 of the manuscript:

By injecting HGO suspension with a spinning nozzle of 500 μm into CaCl₂ aqueous solution, the HGO hydrogel fibers were obtained (Supplementary Fig. 4)...Additionally, in view of the aging effect on the preparation of aerogel materials, the specific surface area, electrical conductivity, tensile strength, and average diameter of HGAFs with different aging times (1, 2, and 3 days) were investigated in Supplementary Figures 10-13. The specific surface area decreases from 231.7 m² g⁻¹ to 162.9 m² g⁻¹ with increasing aging time (Supplementary Figure 10). It can be attributed to the fact that the diffusion of ions during

aging increases the ionic crosslinks between holey graphene sheets, making them more tightly packed and causing more shrinkage (Supplementary Figure 13, the average diameter of the fibers decreases from 329.41 μm to 273.44 μm with aging time). Nevertheless, at the same time, the electrical conductivity (from 146.77 S m^{-1} to 211.69 S m^{-1} , Supplementary Figure 11) and tensile strength (from 0.77 MPa to 1.03 MPa, Supplementary Figure 12) of the fibers increase with the aging time.

Supplementary Figure 10. N₂ sorption isotherms of HGAFs without aging and after aging for 1, 2, and 3 days.

Supplementary Figure 11. Electrical conductivity of HGAFs without aging and after aging for 1, 2, and 3 days.

Supplementary Figure 12. Stress-strain curves of HGAFs without aging and after aging for 1, 2, and 3 days.

Supplementary Figure 13. The average diameter of HGAF without aging and after aging for 1, 2, and 3 days.

On pages 13-14 of the manuscript:

LiCl@HGAFs possess both high solar-thermal conversion and high electro-thermal capability... The overall cycle stability of GAFs and HAGFs is maintained at a relatively stable level up to 10 cycles, both demonstrating rapid cycling capability of water capture and water release (Fig. 2h, Supplementary Fig.22).

Supplementary Figure 22. Cycling stability of the sorption-desorption process of a) LiCl@GAFs and b) LiCl@HGAFs.

Comment 2: “Provide TEM images of LiCl@GAFs and LiCl@HGAFs after cyclic test and compare those with the TEM images of the synthesized LiCl@GAFs and LiCl@HGAFs before any experiment.”

Reply: We thank the reviewer for the valuable suggestions. We have provided the TEM images LiCl@GAFs and LiCl@HGAFs after 10 cycles and compared those with the TEM images of the synthesized LiCl@GAFs and LiCl@HGAFs before any experiment, as shown in **Supplementary Fig. 26-27** in the revised Supplementary Information as well as bellow.

Revision: The following sentence were added on page 14 of the manuscript:

LiCl@HGAFs possess both high solar-thermal conversion and high electro-thermal capability...The obtained transmission electron

microscopy (TEM) images of LiCl@GAFs and LiCl@HGAFs before and after the 10 sorption-desorption cycles show that there is no obvious change in fiber structure before and after the cyclic test (Supplementary Fig. 24). The corresponding elemental maps of LiCl@HGAFs after cyclic test show that Cl element derived from LiCl was uniformly distributed along the holey graphene sheets along with C element, further revealing the satisfactory stability of LiCl@HGAFs (Supplementary Fig. 25).

Supplementary Figure 24. TEM images of LiCl@GAF: a) before the cyclic test and b) after the cyclic test, and TEM images of LiCl@HGAF: c) LiCl@HGAF before the cyclic test and d) after 10 sorption-desorption cycles.

Supplementary Figure 25. TEM images of LiCl@HGAF after the cyclic test and corresponding elemental maps. c) C element and d) Cl element.

Comment 3: “Fit the water uptake data with Freundlich and S-B-K models. Consult the following papers:

R.H. Mohammed et al., Revisiting the adsorption equilibrium equations of silica-gel/water for adsorption cooling applications, *International Journal of Refrigeration* 86 (2018) 40–47.

B.B. Saha et al., Computer simulation of a silica gel water adsorption refrigeration cycle - the influence of operating conditions on cooling output and COP, *ASHRAE Transactions*, Vol. 101, No. 2, pp. 348-357, 1995.”

Reply: We appreciate the reviewer for this suggestion. It is essential to study the adsorption equilibrium model of adsorbents for their

performance in a heat allocation system. We have fitted the water uptake data of sorption isotherm with both the Freundlich and S-B-K models at moderate relative pressure from 0.15 to 0.7. And we have added the fitting results in the Supplementary Information.

Revision: The following sentences were added on pages 15-16 of the manuscript:

As a multifunctional hygroscopic material, in addition to obtaining water from the air, it can also be used as a thermal energy storage material along with an excellent water sorption property.... The isotherm curves obtained at four different temperatures (293 K, 303 K, 313 K, and 323 K) are nearly linear at moderate pressures from 0.15-0.7 and can therefore be described by the Freundlich model and S-B-K model (Supplementary Fig.26 and Supplementary Table 2-3).³⁵⁻³⁶

Supplementary Figure 26. Data fitting of the experimental water sorption isotherm of LiCl@HGAFs-7 by the a) Freundlich equation and b) S-B-K equation.

Supplementary Table 2. Coefficients of the Freundlich equation for LiCl@HGAFs-7

	293 K	303 K	313 K	323 K
X_0	5.013	3.783	3.249	2.842
n	1.099	1.116	1.122	1.226

Supplementary Table 3. Coefficients of the S-B-K equation for LiCl@HGAFs

Parameters	Value
A0	$3.1119 \cdot 10^3$
A1	-29.657
A2	$9.4463 \cdot 10^{-2}$
A3	$-1.0047 \cdot 10^{-4}$
B0	$-2.0222 \cdot 10^2$
B1	1.9929
B2	$-6.5048 \cdot 10^{-3}$
B3	$7.06 \cdot 10^{-6}$

We appreciate the reviewer for the constructive comments and suggestions to improve our manuscript, thanks again!

For Reviewer #3:

We appreciate the reviewer for commenting “*This work reports hygroscopic holey graphene aerogel fibers for highly efficient moisture capture, heat allocation and microwave sorption. A series of experiments were conducted to support the conclusions and claims.*” Regarding the novelty and the raised points, we have addressed all the issues in a point-by-point manner and all the changes to the original manuscript are highlighted in blue.

As for the novelty, we highlight the following five innovative aspects of this work:

- (1) For the first time, the hygroscopic holey graphene aerogel fibers with high sorption capacity, fast moisture capture rate, and superior recyclability are prepared for water capture from the air in this manuscript. This is the first water harvesting material based on flexible aerogel fibers. The hygroscopic holey graphene aerogel fibers exhibit an excellent moisture absorption capacity over 4.15 g g^{-1} at $25 \text{ }^\circ\text{C}$ and 90% relative humidity, fast moisture capture rate of 1.81 g g^{-1} in 30 min, and the moisture absorption capacity retains 95.5% of the initial capacity after 10 cycles.
- (2) Compared with graphene aerogel fibers, the etched nanopores of holey graphene aerogel fibers provide abundant water transport

pathways, which is conducive to improving the mass transfer characteristics of materials.

(3) Different from other moisture sorption materials, the hygroscopic holey graphene aerogel fibers can be regenerated using both photothermal and electrothermal methods with little energy input. The macroscopic one-dimensional conductive material can be easily applied voltage to achieve electrothermal regeneration. Combined with its great adsorption capacity and fast kinetics, it is possible to carry out multiple water harvesting cycles at night.

(4) In this work, we also probe the potential use of hygroscopic holey graphene aerogel fibers in adsorptive heat transfer devices. With water as the working fluid, adsorptive heat transfer and adsorptive clean water production can be integrated to produce cooling and heating energy.

(5) With confined liquid in the confined space, the hygroscopic holey graphene aerogel fibers exhibit intelligent microwave absorption with a broad bandwidth, good impedance matching, and a high attenuation constant. Moreover, the microwave absorption performance of hygroscopic holey graphene aerogel fibers can be adjusted by controlling the water content.

(6) To further demonstrate the novelty, we have also supplemented extra experiments. For instance, we have studied the aging effect of the fibers (Supplementary Figures 10-13), the cycling performance of both LiCl@GAFs and LiCl@HGAFs (Supplementary Figure 22), the dehumidification performance in comparison to other sorbents (Supplementary Figure 19), and the microwave sorption stability (Supplementary Figure 34).

Comment 1: “For moisture sorption, sorption capacity defined as weight of sorbed water/weight of sorbent was adopted to depict the water sorption capacity. Compared with other sorbents, the author claimed that the holey graphene aerogel have the highest sorption capacity. However, the excellent sorption capacity of the fiber is mainly due to its low density, which is not available in real applications. I suggest that the author supplement the following experiments: various moisture sorbents with the same weight or volume are put into a closed space with the same initial relative humidity; after a period of time, to evaluate which sorbent reduces the relative humidity most.”

Reply: We thank the reviewer for this comment. According to the suggestions, the results of the dehumidification experiment have been provided. At room temperature, a series of 2 g and packing volume of 5 cm³ dehumidifying materials were respectively placed in a sealed chamber with a size of 60*50*50 cm and an initial relative humidity of 90%, and the relative humidity in the chamber was detected after 6 hours. For

the dehumidification test of moisture sorption materials with the same mass (Supplementary Fig. 19a), LiCl@HGAFs outperforms most of the compared sorbents and are slightly weaker than LiCl. For the dehumidification test of moisture sorption materials with the same volume (Supplementary Fig. 19b), LiCl@HGAFs shows moderate hygroscopic performance compared with other sorption materials. However, LiCl@HGAFs, as new and flexible macro fibrous hygroscopic materials, have a more flexible adsorption-desorption cycle compared with other hygroscopic materials reported in the literature. In air-water intake applications, hygroscopic materials are typically set to adsorb at night and are regenerated during the daytime through photothermal processes. LiCl@HGAFs, on the other hand, not only can be desorbed by photothermal energy in the daytime, but also can be electrically desorbed at night, which means it can perform more adsorption-desorption cycles over a whole day. Therefore, the hygroscopic holey graphene aerogels in our work would be of great benefit for water harvest and the multifunctional materials are greatly desirable in practical applications.

Revision: The following content has been added in the manuscript on page 12:

Combining the various pore structure and high porosity of the aerogel matrix with the strong moisture sorption of LiCl, LiCl@HGAFs

show excellent moisture sorption capacity... Furthermore, dehumidification tests of a series of moisture sorption materials were conducted in a closed chamber with the same initial relative humidity (Supplementary Fig.19). For the dehumidification performance of the sorbents with the same mass (Supplementary Fig. 19a), LiCl@HGAFs outperforms most of the compared sorbents and is slightly weaker than LiCl. For the sorbents with the same volume (Supplementary Fig. 19b), LiCl@HGAFs shows moderate hygroscopic performance compared with other sorption materials.

Supplementary Figure 19. Dehumidification performance of LiCl@HGAFs compared with other moisture sorption materials in a) the same mass (2 g) and b) the same packing volume (5 cm³). The moisture sorption materials include commercial color-changing silical gel (methyl violet@silical gel), commercial hygroscopic fibers (EKS fibers from

TOYOBO CO., LTD), LiCl, UiO-66 (zirconium 1,4-dicarboxybenzene MOF), LiCl@UiO-66, active carbon fiber loaded with lithium chloride (LiCl@ACF), and LiCl@HGAFs. The masses of moisture sorption materials with the same volume in b) are 3.55 g, 2.92 g, 2.74 g, 3.12 g, 2.84 g, and 1.76 g, respectively.

Comment 2: “For microwave sorption, this work show that the fiber possess good microwave sorption own to the entrapped water. Therefore, an experiment about the sorption stability of microwave should be supplemented in that the water will be heated after microwave sorption, which will inevitably lead to the decrease of sorption performance.”

Reply: We appreciate the reviewer for the comment on the sorption stability of the microwave. During the irradiation of microwave, the water content entrapped in the fibers should be a concern under a high electromagnetic power because of the loss of water. However, under a mild or low electromagnetic power, the fibers can well retain the water since the LiCl@HGAF-H₂O sample is sealed in the paraffin during the testing with the network analyzer, and therefore show good stability for microwave sorption. Furthermore, we are more concerned with the adjustable microwave absorption properties of materials. Once the traditional microwave absorption materials are prepared, their microwave sorption properties cannot be modulated. In our work, the microwave

absorption properties can be tuned by adjusting the water sorption content and tuning the power of the external stimuli (solar energy or electric field). Based on the reviewer's suggestion, we have supplemented the experiment about the sorption stability of microwaves. We placed the LiCl@HGAF-H₂O fibers in the testing chamber and set the test power to +10 dBm (10 mW) to keep the materials in the electromagnetic environment all the time. After that, this test was conducted every 2 h to obtain the changes in the microwave absorption properties of the materials.

Revision:

According to equation S8 and equation S9 and the transmission line theory, the electromagnetic wave absorption properties (reflection loss, RL) of different HGAF profiles were calculated...Since the microwave sorption stability of hygroscopic holey graphene aerogel fibers containing water is crucial in real applications, we characterized the long-time stability of LiCl@HGAF-H₂O fibers in an electromagnetic environment with the power of +10 dBm. The samples with variant thicknesses exhibit stable microwave sorption behavior up to 12 h (Supplementary Fig. 33a~f). The maximum absorptivity of the fibers was above 99% invariably (Supplementary Fig. 33g).

Supplementary Figure 33. The sorption stability of microwave test for LiCl@HGAF-H₂O at different thicknesses. The samples were placed in

the electromagnetic environment (10 dBm) for a) 2 h, b) 4 h, c) 6 h, d) 8 h, e) 10 h, and f) 12 h, respectively. g) Maximum absorptivity (%) with the thickness of 2.0 mm with different radiation times.

We appreciate the reviewer for the constructive comments and suggestions to improve our manuscript, thanks again!

REVIEWERS' COMMENTS

Reviewer #1 (Remarks to the Author):

The authors have well revised manuscript according to the reviewers' comments. The reviewer suggests that this paper should be accepted.

Reviewer #2 (Remarks to the Author):

The authors provide a feasible method for fabricating hygroscopic holey graphene aerogel fibers with integrated moisture capture, heat allocation, and microwave absorption characteristics. Due to the super high surface area and strong water uptake kinetics for water transport, the produced hygroscopic holey graphene aerogel fibers have a water sorption capacity of around 4.15 g g⁻¹ at p/p₀=0.9, outperforming traditional moisture adsorbents. The work is novel, and it is inspiring. I am suggesting the acceptance of the paper.

We appreciate all the reviewers for their hard work and constructive comments.